# Rank orders and signed interactions in evolutionary biology

Kristina Crona*

Mathematics and Statistics, American University, Washington DC, United States

**Abstract** Rank orders have been studied in evolutionary biology for almost a hundred years. Constraints on the order in which mutations accumulate are known from cancer drug treatment, and order constraints for species invasions are important in ecology. However, current theory on rank orders in biology is somewhat fragmented. Here, we show how our previous work on inferring genetic interactions from comparative fitness data (Crona et al., 2017) is related to an influential approach to rank orders based on sign epistasis. Our approach depends on order perturbations that indicate interactions. We apply our results to malaria parasites and find that order perturbations beyond sign epistasis are prevalent in the antimalarial drug-resistance landscape. This finding agrees with the observation that reversed evolution back to the ancestral type is difficult. Another application concerns the adaptation of bacteria to a methanol environment.

**\*For correspondence:**
kcrona@american.edu

**Competing interests:** The author declares that no competing interests exist.

## Introduction

Rank orders of genotypes with respect to fitness are central in evolutionary biology. Key concepts, such as peaks and mutational trajectories, refer to an underlying rank order. The impact of suboptimal peaks has been discussed ever since fitness landscapes were introduced (*Wright, 1932*). A major advantage with rank orders, as compared to other statistics for fitness landscape, is that order data is robust for variation in experimental methods and measurements. In fact, one does not necessarily need to measure fitness at all. If a rank order can be established by a competition experiment or by any type of rank order preserving fitness proxy, then one can apply theory on rank orders. Results from different empirical studies can therefore be compared; fitness or log-fitness, radius or area, change per minute or generation, makes no difference if rank orders are considered.

The concept of sign epistasis introduced by *Weinreich et al. (2005)*, has been important for recent work on rank orders in evolutionary biology (*Kaznatcheev, 2019*; *Crona et al., 2017*; *Wu et al., 2016*; *de Visser and Krug, 2014*; *Crona et al., 2013*; *Poelwijk et al., 2011*). A system has sign epistasis if the sign of the effect of a mutation, whether positive or negative, depends on genetic background. The graphs in *Figure 1* illustrate the absence of sign epistasis (A), sign epistasis (B), and reciprocal sign epistasis (C) for two-locus systems. Each arrow points toward the genotype of higher fitness. The importance of rank order data for two-locus subsystems is that they carry information about global aspects of fitness landscapes, and therefore about long-term evolution. For instance, all multi-peaked fitness landscapes have reciprocal sign epistasis (*Poelwijk et al., 2011*). Conversely, if subsystems of type C exist, but no systems of type B, then the landscape is multi-peaked (*Crona et al., 2013*).

The squares representing a two-locus subsystem (see the orange square in *Figure 2*) are clearly informative. However larger rectangles can improve the precision of the analysis. Each arrow in *Figure 2* points toward the genotype of higher fitness. The undirected edges connect mutational neighbors, that is genotypes that differ at one locus only. The blue rectangles in *Figure 2* concern replacements of the type $00 \mapsto 11$ for the second and third loci. If the sign of the effect of such a replacement depends on background, then the long arrows have different directions (*Figure 2*B and D). Such a perturbation is similar to sign epistasis, except that two loci are replaced.

We propose a broader perspective on perturbations, that takes advantage of all rectangles in the cube (or hypercube). Differently expressed, in addition to single mutations it is sometimes useful to consider the effects of double mutations and any higher order mutations, as well as replacements $0 \mapsto 1$ and $1 \mapsto 0$ for any selected subset of loci. If the sign of the effect of such a replacement depends on background, then the system has a rectangular perturbation. As proof of principle, we demonstrate that the large rectangles give new insights for two empirical studies (as compared to an analysis of sign epistasis). The first concerns antimalarial drug resistance (*Ogbunugafor and Hartl, 2016*), whereas the second example involves bacteria adapting to a methanol environment (*Chou et al., 2011*).

As in *Crona et al. (2017)*, fitness is additive if the fitness effects of individual mutations sum, otherwise the system has epistasis (or gene interactions). By a rank order induced, or signed, interaction, we mean that the rank order reveals epistasis. Questions about how different approaches to signed interactions relate and how it is possible to determine whether a rank order is compatible with additive fitness were briefly discussed by the authors. As eLife reviewers pointed out, the signed interactions in *Crona et al. (2017)* are not generalizations of sign epistasis but there is clearly some connection. The remaining open problems were the starting point for this work. The main idea for resolving the problems was to consider rectangles such as those in *Figure 2* (no knowledge of Walsh-coefficient, polytopes, or similarly is assumed here).

## Results

We consider biallelic $n$-locus systems. The fitness of a genotype $g$ is denoted $w_g$. For simplicity, we assume that there exists a total order of the genotypes with respect to fitness, here referred to as a rank order (no two genotypes have equal fitness).

For two-locus systems, the expression $u = w_{00} + w_{11} - w_{10} - w_{01}$ measures epistasis. If it easy to verify that $u = 0$ if fitness is additive. By definition, the system has an interaction if $u \neq 0$, and a signed interaction if the rank order implies that $u > 0$ or $u < 0$. For instance the order $w_{11} > w_{00} > w_{10} > w_{01}$ implies that $u > 0$.

For a systematic treatment of order perturbations, it is necessary to write the information in compact form. Notice that a rank order where $w_{00} > w_{10}$ and $w_{01} < w_{11}$ implies sign epistasis, as the effect of a mutation at the first locus can increase or decrease fitness. Reciprocal sign epistasis for a two-locus system can be defined as a system with two peaks (*Figure 1C*). In particular, parallel arrows point in different directions.

It is straightforward to a check that a two-locus system has sign epistasis exactly if at least one of the following expressions is negative, and reciprocal sign epistasis if both of them are negative (*Figure 1*).

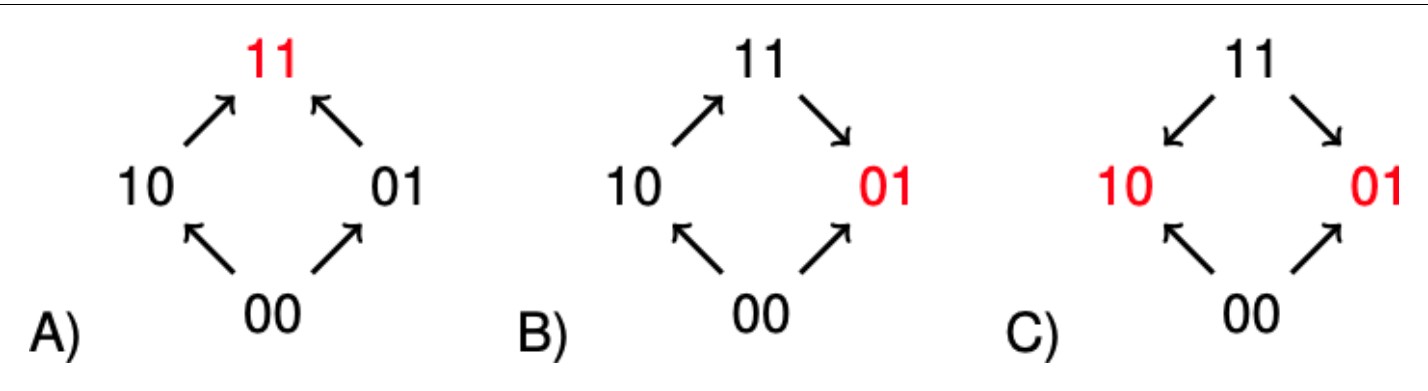

**Figure 1.** The graphs illustrate (**A**) no sign epistasis, (**B**) sign epistasis, and (**C**) reciprocal sign epistasis. The peaks are marked red. Under the assumption that 00 has minimal fitness, and that the genotypes are positioned as in the figure, the three types can be characterized as graphs with no arrows down, one arrow down, or two arrows down.

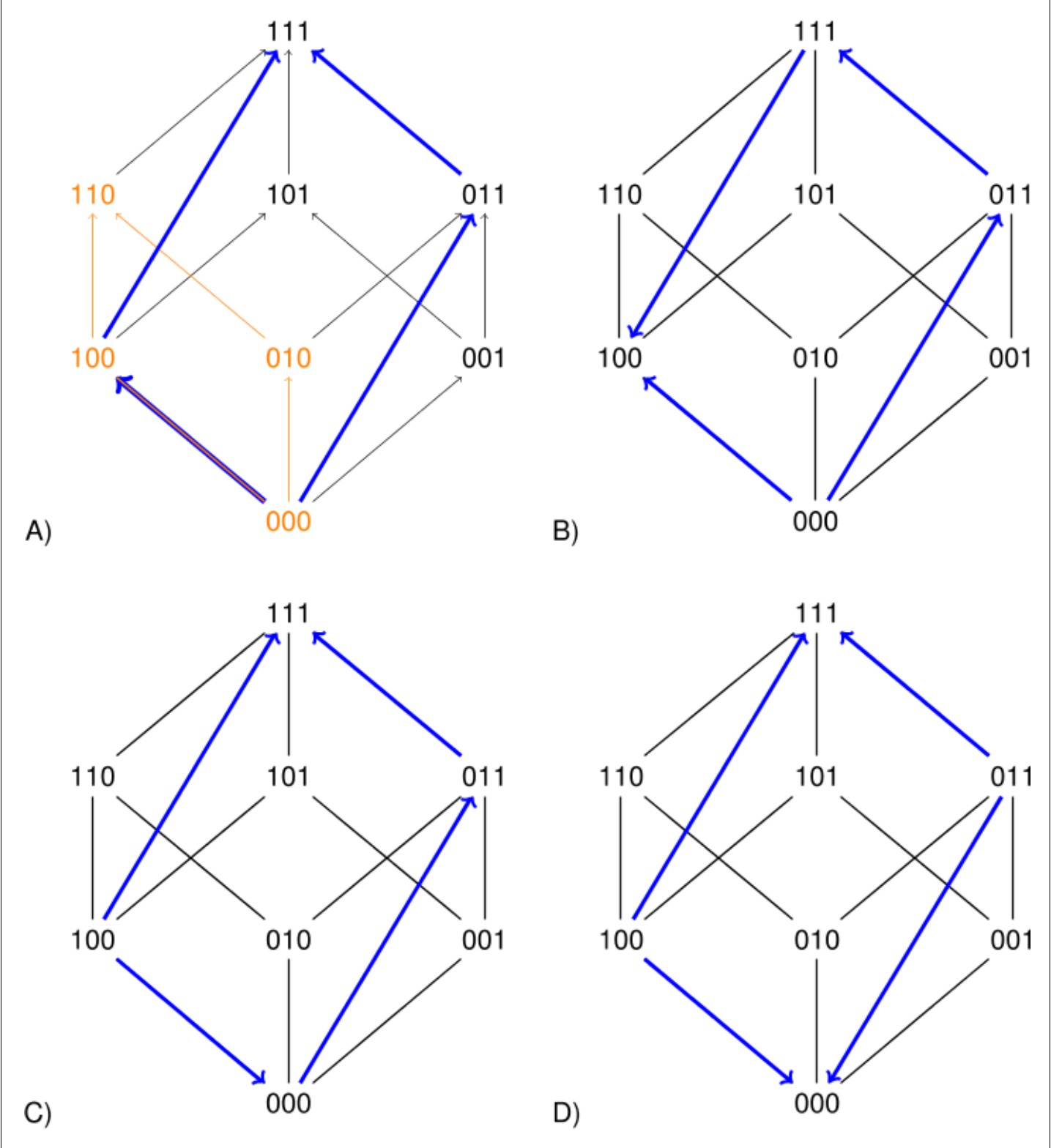

**Figure 2.** Each arrow points toward the genotype of higher fitness. The undirected edges connect mutational neighbors and carry no information about fitness differences. The graph 2A is compatible with additive fitness. The other graphs are not compatible with additive fitness because at least one pair of parallel arrows point in different directions. The lower graphs indicate sign epistasis, as is clear from the short arrows. The right graphs indicate size two perturbations, as is clear from the long arrows.

$$(w_{00} - w_{10})(w_{01} - w_{11}) \tag{1}$$

$$(w_{00} - w_{01})(w_{10} - w_{11}) \tag{2}$$

A negative sign corresponds to that a pair of parallel arrows that point in different directions. Large systems can be analyzed similarly.

A system has a rectangular perturbation if the sign of the effect of replacing a subset of loci, according to the rule $0 \mapsto 1$ and $1 \mapsto 0$, depends on background. The size of the perturbation refers to the number of loci replaced.

In particular, a rectangular perturbation of size 1 is a case of sign epistasis. The expression $u$ was used to analyze gene interactions in the two-locus case, and a similar approach works in general. We define $\mathcal{C}$ as the set of all expressions of the type

$$r = w_g + w_{g'} - w_{g''} - w_{g'''},$$

for genotypes $g, g', g'', g'''$ such that $r = 0$ if fitness is additive. For instance, for $n = 3$, the expressions

$$w_{110} + w_{000} - w_{100} - w_{010} \text{ and } w_{111} + w_{000} - w_{100} - w_{011}$$

associated with two of the rectangles in *Figure 2* (i.e., the leftmost side and the blue rectangle) are included in $\mathcal{C}$. Note that each expression in $\mathcal{C}$ can be obtained by going around a rectangle in the cube and assigning coefficients with alternating signs along the way (each $w$-coefficient is +1 or –1).

We call the elements in $\mathcal{C}$ circuits, with reference to theory presented in *Beerenwinkel et al. (2007b)* (see also 'Materials and methods'; no knowledge of circuits is assumed). By a rank-order-induced circuit, or signed circuit, we mean that the rank order implies that the circuit is positive or negative (exactly as in the two-locus case).

For $n = 3$, consider the circuit $w_{111} + w_{000} - w_{100} - w_{011}$ and the related expressions (the four genotypes from the circuit can be combined in two different ways):

$$(w_{000} - w_{100})(w_{011} - w_{111}) \tag{3}$$

$$(w_{000} - w_{011})(w_{100} - w_{111}). \tag{4}$$

If the rank order implies that expression (3) is negative, then the system has sign epistasis, and if expression (4) is negative, the system has an order perturbation of size 2. The short blue arrows in *Figure 2* point in different directions in the first case, and the long blue arrows do so in the latter. Similarly, any rectangular perturbation corresponds to two parallel arrows (along opposite sides of a rectangle) that point in different directions.

**Remark 1**. The relation between signed circuits and sign epistasis can be summarized as follows for $n = 2$.

 i. A rank order of the genotypes 00, 10, 01 and 11 with respect to fitness implies that $u > 0$ or $u < 0$ exactly if the system has sign epistasis.

 ii. From the information that the rank order implies $u > 0$ (or $u < 0$) alone, one cannot determine whether there are one or two order perturbations, that is, whether or not the system has reciprocal sign epistasis.

 iii. The system has reciprocal sign epistasis if both expressions (1) and (2) are negative, and sign epistasis if at least one of them is negative.

**Remark 2**. A similar observation holds for signed circuits and any $n$.

 i. Each case of sign epistasis is associated with a signed interaction for a circuit in $\mathcal{C}$, that is the rank order of genotypes with respect to fitness implies that the form is positive or negative.

 ii. More generally, each rectangular perturbation is associated with a signed circuit interaction for a circuit in $\mathcal{C}$.

 iii. Each circuit in $\mathcal{C}$ corresponds exactly to two potential rectangular perturbations.

**Table 1.** Rectangular perturbations for drug-exposed and drug-free malaria fitness landscapes. The third line shows the total number of expressions checked. The prevalence of sign epistasis is similar for the landscapes, whereas the remaining perturbations differ by a factor of two.

| Perturbation size | 1 | 2 | 3 | Size (1-3) |
|---|---|---|---|---|
| Drug-exposed | 55 | 21 | 5 | 81 |
| Drug-free | 54 | 39 | 9 | 102 |
| Expressions checked | 112 | 72 | 16 | 200 |

In particular, in order to check a three-locus system for perturbations it is sufficient to check the signs of 24 expressions, including expression (3) and (4), associated with 12 circuits (see 'Materials and methods').

## Applications

The first application concerns a study of the malaria-causing parasite *Plasmodium vivax* (*Ogbunugafor and Hartl, 2016*). The original study investigates a four-locus system exposed to different concentrations of the anti-malarial drug pyrimethamine (PYR). The quadruple mutant denoted 1111 has the highest degree of drug resistance, whereas the genotype 0000 has the highest fitness among all genotypes in the drug-free environment. Several concentrations of the drug were tested in the original study. We compared the highest concentration of the drug and the drug-free environment. Sign epistasis was equally frequent in both fitness landscapes. However, the drug-free environment had about twice as many perturbations of size 2 and 3, as the drug-exposed environment. The summary statistic indicates that (*Table 1*) adaptation is more difficult in the drug-free environment, because few replacements of pieces of code (regardless of size) are universally beneficial.

This finding agrees well with the authors' observation that resistance development is a relatively straightforward process, whereas reversed evolution from the mutant 1111 back to the ancestral type is difficult. Interestingly, the complete statistics of order perturbations was better able to distinguish the landscapes than sign epistasis alone.

The second application concerns a study of the bacterium *Methylobacterium extorquens*, which adapts to a methanol environment (*Chou et al., 2011*). For the sake of the argument, we assume that the published measurements are exact (a discussion about the impact of measurement errors can be found in 'Materials and methods'). The fitness landscape does not have sign epistasis. However, the system has size 2 perturbations (*Table 2*). In particular $w_{1000} > w_{0100}$ and $w_{1011} < w_{0111}$, so that the sign of the effect of the change $10 \mapsto 01$ at the first pair of loci depends on background. The perturbations show that the rank order is incompatible with additive fitness.

A complete analysis of perturbations for an $n$-locus system requires an investigation of $2\left(\frac{6^n}{8} - 4^{n-1} + 2^{n-3}\right)$ expressions for $n \geq 3$ (Theorem 1, 'Materials and methods'). Statistics on order perturbations should always be considered estimates because of measurement errors. However, that is true for all rank-order-based concepts, and a peak count is sensitive to noise. One could naively believe that a broader perspective makes the problem worse. By contrast, rectangular perturbations can sometimes be helpful in separating signal and noise.

For instance, assume that fitness is additive and that all mutations decrease fitness (*Figure 3*). The graphs show the proportion of unexpected observations, that is increased fitness, for replacements of the type $0 \mapsto 1$, $00 \mapsto 11$ and $000 \mapsto 111$ for two error distributions (errors follow a normal distribution in the upper graph and the Student's t-distribution with df = 3 in the lower, see 'Materials and methods' for more details). In both cases, the proportion of unexpected results decreases by the number of loci replaced. The reason is that the noise level is similar in all cases, whereas the effect size is not.

This observation can be useful. For instance, it is difficult to exclude additive fitness for a collection of detrimental mutations by observing sign epistasis alone (apparent sign epistasis can result from noise). However, an investigation of the three types of replacements described could be conclusive. In particular, similar proportions of unexpected results would constitute a strong argument against additive fitness.

Rank orders and perturbations are important for problems beyond natural selection. Constraints on the orders in which mutation occur are known for development of cancer drug resistance (*Hosseini et al., 2019*; *Beerenwinkel et al., 2007a*). In brief, a typical case would be that some mutation $B$ is not selected for unless a mutation $A$ has occurred, even though $AB$ has high fitness. If the constraint holds universally, it can be described as: $w_{01*} < w_{00*} < w_{10*} < w_{11*}$, where $*$ is an arbitrary sequence. For an $n$-locus system, such a constraint implies $2^{n-2}$ order perturbations if fitness is additive for the remaining loci.

## Additivity and rank orders

The problem of determining if rank orders are compatible with additive fitness is already interesting for $n = 3$.

For the fitness landscape

$$w_{000} = 1, w_{100} = 1.1, w_{010} = 1.12, w_{001} = 1.09,$$

$$w_{110} = 1.2, w_{101} = 1.22, w_{011} = 1.19, w_{111} = 1.3,$$

the rank order is $w_{111} > w_{101} > w_{110} > w_{011} > w_{010} > w_{100} > w_{001} > w_{000}$. The change $0 \mapsto 1$ at any locus increases fitness regardless of background, so there is no sign epistasis. However, the sign of the effect of the change $10 \mapsto 01$ at the first pair of loci depends on background, as $w_{100} < w_{010}$ and $w_{101} > w_{011}$. In particular, the rank order is not compatible with additive fitness.

As remarked in *Crona et al. (2017)*, exactly 384 orders are compatible with the absence of sign epistasis for $n = 3$. By applying order perturbations (and by inspection), we verified that exactly 96 out of the 384 orders are compatible with additive fitness, or 0.24 percent of all 40,320 rank orders for three-locus systems. After relabeling (see 'Materials and methods' for details), only the following two orders are compatible with additive fitness:

$$w_{111} > w_{110} > w_{101} > w_{011} > w_{100} > w_{010} > w_{001} > w_{000}$$

$$w_{111} > w_{110} > w_{101} > w_{100} > w_{011} > w_{010} > w_{001} > w_{000}.$$

Theory on rank orders and additivity has been developed independent of biological application and is still an active research area (*Searles and Slinko, 2015*; *Maclagan, 1998*). In principle, one can determine whether a specific rank order is compatible with additive fitness by solving a system of linear inequalities obtained from the order. It is straightforward to verify that a rank order is compatible with additive fitness unless it has rectangular perturbations for $n$=4. However, counterexamples to the analogous statement for $n = 5$ exist (*Kraft et al., 1959*) (see Materials and methods for more details).

## Discussion

A rank order of genotypes, from highest to lowest fitness, is informative about gene interactions (*Crona et al., 2017*). We introduced rectangular perturbations to resolve problems on how different approaches to rank order induced (or signed) interactions relate. Sign epistasis concerns the effects of single mutations, and the new perturbations also concern the effects of multiple mutations.

We have established that sign epistasis implies that a system has signed circuit interactions, but the converse is not true. Strictly speaking, signed circuits are not generalizations of sign epistasis, neither are signed circuits refinements of sign epistasis. However, sign epistasis can be seen as rectangular perturbations of minimal size.

A rank order is compatible with additive fitness unless there is sign epistasis for a two-locus system. We have provided a counterexample for the analogous claim for three loci. For four-locus systems, a rank order is compatible with additive fitness unless there are rectangular perturbations. However, counterexamples exist for five loci (*Maclagan, 1998*; *Kraft et al., 1959*). In general, only a very small proportion of all rank orders are compatible with additive fitness, and the theoretical understanding for the property is limited.

As a proof of principle, we applied rectangular perturbations to empirical studies on antimalarial drug resistance (*Ogbunugafor and Hartl, 2016*) and on bacteria adapting to a methanol

environment (*Chou et al., 2011*). Rectangular perturbations have the capacity to detect epistasis when conventional rank-order-based methods cannot. The perturbations capture evolutionary important properties beyond local aspects. A complete analysis of rectangular perturbations requires an investigation of a large number of expressions (of the order $6^n$). Often, it is meaningful to consider selected perturbations, depending on the context.

In general, rank orders are quite informative about gene interactions, and also regarding evolutionary potential (in a qualitative sense) provided one assumes the Strong Selection Weak Mutation regime (*Gillespie, 1984*). If the available information from an empirical study is a rank order, for instance from a competition experiment, then it is obviously useful to have methods for interpreting rank orders (further motivations are discussed in *Crona et al., 2017*). However, rank-orders methods have obvious limitations. Rank orders are insufficient for determining the effect of genetic recombination (*de Visser et al., 2009*; *Otto and Lenormand, 2002*). Evolutionary predictability is sensitive for population parameters, and the importance of accessible mutational trajectories is not universal (*de Visser and Krug, 2014*; *Krug, 2019*).

It would be interesting to determine the extent to which rectangular perturbations are helpful for relating local and global properties of fitness landscapes, similar to results on sign epistasis (*Weinreich et al., 2005*; *Poelwijk et al., 2011*; *Crona et al., 2013*), and also in analyzing statistical aspects further. Algorithms and exact formulas for potential perturbations are provided in the 'Materials and methods'.

## Materials and methods

### General circuits

For a biallelic $n$-locus system, we have discussed a set $\mathcal{C}$ defined as all expressions of the form

$$r = w_g + w_{g'} - w_{g''} - w_{g'''},$$

for genotypes $g, g', g'', g'''$, such that $r = 0$ if fitness is additive. In order to connect to general theory, we need some concepts. Starting with the two-locus case, for the genotypes $00, 10, 01, 11$, one can form vectors in $\mathbb{R}^3$ by adding an extra coordinate one for each genotype

$$(0,0,1), (1,0,1), (0,1,1), (1,1,1).$$

The four vectors are linearly dependent since

$$(1,1,1) + (0,0,1) - (1,0,1) - (0,1,1) = (0,0,0).$$

The dependence relation corresponds exactly to the linear form

$$u = w_{11} + w_{00} - w_{10} - w_{01}.$$

Similarly, one can form vectors in $\mathbb{R}^{n+1}$ from the vertices of the $n$-cube by adding a coordinate 1. Circuits are defined as the *minimal dependence relations*, in the sense that each proper subset of the vectors (with non-zero coefficients) are linearly independent.

There are in total 20 circuits for $n = 3$, including $\mathcal{C}$, but also for instance the circuit

$$w_{000} + 2w_{111} - w_{110} - w_{101} - w_{011}.$$

In this terminology, one can describe $\mathcal{C}$ as the set of circuits that have exactly four non-zero coefficients for the variables $w_g$. Note that all circuits are zero if fitness is additive. In that sense, circuits

**Table 2.** Order perturbations of size 1 and 2 for *Methylobacterium extorquens*.
The landscape has no sign epistasis. However, perturbations of size 2 reveal that the landscape is not additive.

| Perturbation of size | 1 | 2 |
|---|---|---|
| Number of perturbations | 0 | 3 |
| Expressions checked | 112 | 72 |

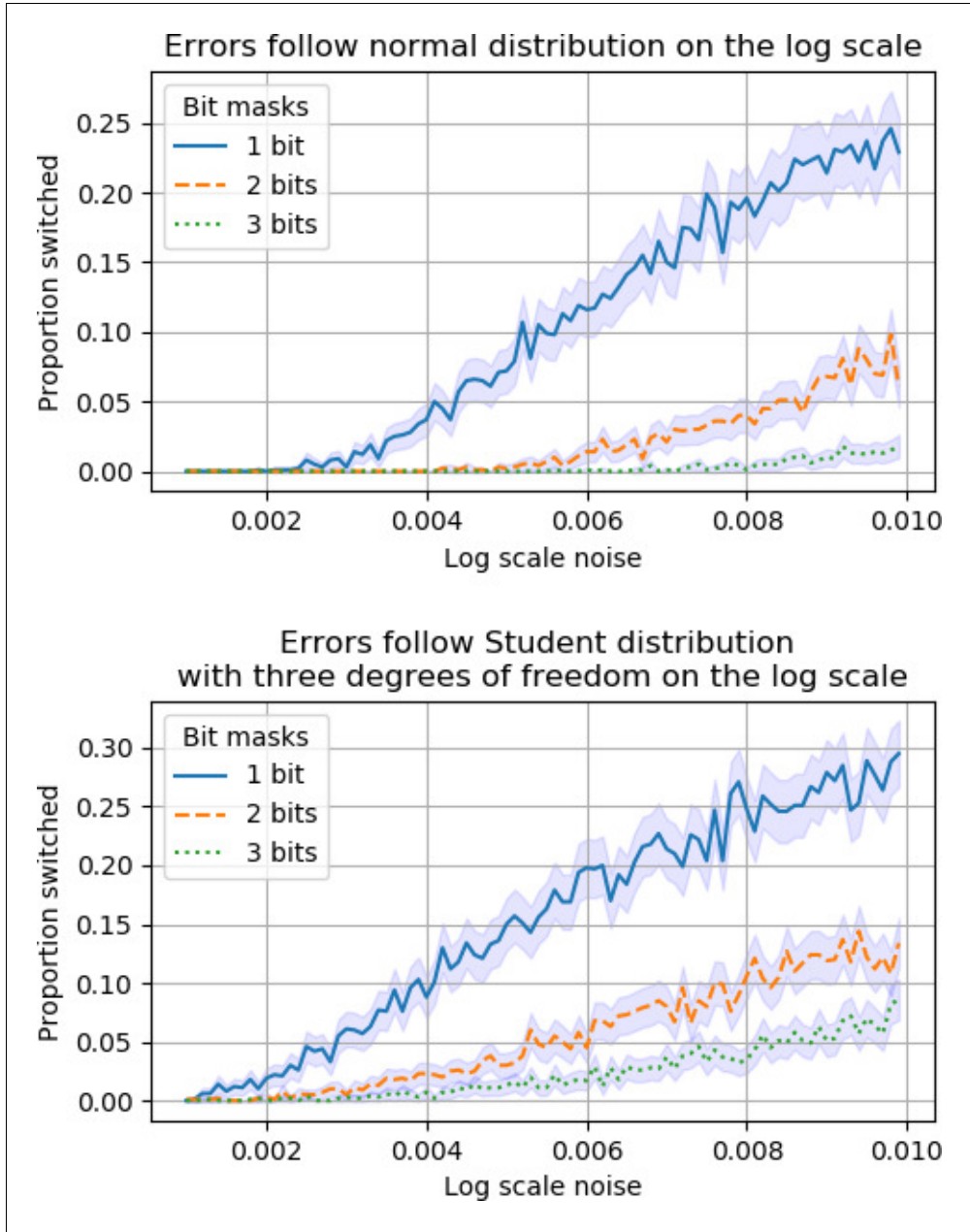

**Figure 3.** The fitness landscapes are additive and each mutation $0 \mapsto 1$ decreases fitness. The graphs compare replacements $0 \mapsto 1, 00 \mapsto 11$ and $000 \mapsto 111$. Because of measurement errors, some replacements appear to increase fitness. The graphs show the proportion of such cases.

measure epistasis. An analysis of all circuits provides very complete information on gene interactions (*Beerenwinkel et al., 2007b*).

## Counting rectangular perturbation

To count rectangular perturbations, one needs to find all rectangles with vertices in an $n$-cube. We provide an explicit formula with proof for the reader's convenience, even though the result is elementary. The proof depends on Stirling numbers of the second kind. The Stirling number $S(n, k)$ is defined as the number of ways to partition a set of $n$ objects into $k$ non-empty subsets. We refer to *Grimaldi (2006)* for more background.

**Theorem 1**. The total number of (potential) rectangular perturbations for an $n$-locus system is

$$2\left(\frac{6^n}{8} - 4^{n-1} + 2^{n-3}\right) \text{ for } n \geq 3.$$

Moreover, the number of rectangular perturbations of size $k$ (exactly $k$ loci are replaced) equals

$$2^{k-1}\binom{n}{k}\binom{2^{n-k}}{2}.$$

**Corollary**. A complete investigation of sign epistasis for an $n$-locus system requires that one checks the signs of $n\binom{2^{n-1}}{2}$ expressions.

**Lemma 1**. For Stirling numbers of the second kind $S(n,k)$, the following identities hold.

$$S(n,2) = 2^{n-1} - 1$$

$$S(n,3) = \frac{1}{6}(3^n - 3 \cdot 2^n + 3).$$

**Proof**. The first formula holds since there are $2^n - 2$ non-empty proper subsets of $n$ elements, and each partition corresponds to exactly two subsets. Similarly, the second formula can be derived from the observation that one can construct three labeled subsets of $n$ elements in $3^n$ ways. After reducing for all cases with empty sets, the number of alternatives is

$$3^n - 3(2^n - 2) - 3 = 3^n - 3 \cdot 2^n + 3.$$

Each partition corresponds to six alternatives, which completes the argument.

**Lemma 2**. Let $n \geq 3$. There are

$$6^n/8 - 4^{n-1} + 2^{n-3}$$

rectangles with vertices in an $n$-cube.

**Proof.** For each vertex $s_1 \ldots s_n$ in an $n$-cube, one can construct a rectangle with vertices on the $n$-cube as follows. Distribute the set of $n$ loci into three subsets $S_1, S_2$ and $S_3$, where the intersection of each pair of sets is empty, and where $S_1$ and $S_2$ are non-empty.

From the vertex $s_1 \ldots s_n$, one constructs the remaining vertices by replacing sets of loci, according to the rule $0 \mapsto 1$, and $1 \mapsto 0$. Specifically, one vertex is obtained by replacing all elements in $S_1$, one by replacing all elements in $S_2$, and the last by replacing all elements in $S_1 \bigcup S_2$. If $S_3$ is empty, one can construct $S(n,2)$ rectangles. If all three sets are non-empty, then one can choose $S_3$ in three ways, and consequently construct $3 \cdot S(n,3)$ rectangles. In total, one obtains $3 \cdot S(n,3) + S(n,2)$ rectangles starting from a particular vertex. There are $2^n$ vertices in the $n$-cube and each rectangle has four vertices. By the previous lemma, the number of rectangles is

$$\frac{2^n}{4} \cdot (3 \cdot S(n,3) + S(n,2)) = 6^n/8 - 4^{n-1} + 2^{n-3},$$

which completes the proof.

We can now prove the main result.

**Proof of Theorem 1**. The total number of (potential) rectangular perturbations for an $n$-locus system is

$$2\left(\frac{6^n}{8} - 4^{n-1} + 2^{n-3}\right).$$

A rectangle with vertices in the $n$-cube corresponds to exactly two rectangular perturbations, one for each pair of parallel edges. Consequently, the result follows from Lemma 2.

The second part of the theorem states that the number of rectangular perturbations where exactly $k$ loci are replaced is equal to

$$2^{k-1}\binom{n}{k}\binom{2^{n-k}}{2}.$$

The positions of the $k$ loci that change can be chosen in $\binom{n}{k}$ different ways. There are $2^k$ words of length $k$, and therefore $2^{k-1}$ pairs consisting of a word and its replacement (for instance, the replacement of 110 is 001). Finally, there are $2^{n-k}$ different backgrounds, so that a pair of backgrounds can be chosen in $\binom{2^{n-k}}{2}$ ways.

As mentioned, a single signed circuit may correspond to two cases of sign epistasis. Indeed, a two-locus system with reciprocal sign epistasis is an example. It is thus of interest to identify all order perturbations, rather than identifying signed circuits only. Theorem 1 and its proof indicate how one can find all order perturbations. In particular, the complete list for identifying rectangular perturbations for $n = 3$ consists of the following 24 expressions, where the first 18 expressions concern sign epistasis, and the remaining six size 2 perturbations.

$$(w_{000} - w_{100})(w_{010} - w_{110}), (w_{000} - w_{100})(w_{001} - w_{101}), (w_{000} - w_{100})(w_{011} - w_{111}),$$

$$(w_{010} - w_{110})(w_{001} - w_{101}), (w_{010} - w_{110})(w_{011} - w_{111}), (w_{001} - w_{101})(w_{011} - w_{111}),$$

$$(w_{000} - w_{010})(w_{100} - w_{110}), (w_{000} - w_{010})(w_{001} - w_{011}), (w_{000} - w_{010})(w_{101} - w_{111}),$$

$$(w_{100} - w_{110})(w_{001} - w_{011}), (w_{100} - w_{110})(w_{101} - w_{111}), (w_{001} - w_{011})(w_{101} - w_{111}),$$

$$(w_{000} - w_{001})(w_{100} - w_{101}), (w_{000} - w_{001})(w_{010} - w_{011}), (w_{000} - w_{001})(w_{110} - w_{111}),$$

$$(w_{100} - w_{101})(w_{010} - w_{011}), (w_{100} - w_{101})(w_{110} - w_{111}), (w_{010} - w_{011})(w_{110} - w_{111}),$$

$$(w_{000} - w_{110})(w_{001} - w_{111}), (w_{000} - w_{101})(w_{010} - w_{111}), (w_{000} - w_{011})(w_{100} - w_{111}),$$

$$(w_{100} - w_{010})(w_{101} - w_{011}), (w_{100} - w_{001})(w_{110} - w_{011}), (w_{010} - w_{001})(w_{110} - w_{101}).$$

## Rank orders compatible with additive fitness

Different rank orders may differ only 'by labels'. For instance, the rank orders

$$w_{11} > w_{10} > w_{01} > w_{00} \text{ and } w_{00} > w_{01} > w_{10} > w_{11}$$

differ by the map $0 \mapsto 1$ and $1 \mapsto 0$, applied to both loci for each genotype. By definition, a cube isomorphism preserves the adjacency structure of the cube (a pair of mutational neighbors is mapped to a pair of neighbors).

Two rank orders are considered equivalent if a cube isomorphism induces a map between them. Differently expressed, for any given rank order (for a two-locus system), one can obtain an equivalent order by assigning the label 00 to a genotype of choice (among four alternatives) and then 10 to one of its neighbors (among two alternatives). After that, the adjacency condition determines the labels of the remaining genotypes. The new rank order is identical to the original one, except that the genotypes have new labels as described. It follows that each equivalence class consists of $8 = 4 \cdot 2$ orders. The 24 rank orders for a two-locus systems can be partitioned into $\frac{24}{8} = 3$ equivalence classes.

For general $n$, each equivalence class consists of $2^n \cdot (n!)$ rank orders by a similar argument. In particular, for $n = 3$, each equivalence class consists of 48 rank orders, and the 8! rank orders can be partitioned into 840 equivalence classes. Questions on rank orders for $n = 3$ are manageable as it suffices to check 840 orders.

To determine whether a specific rank order is compatible with additive fitness, one has to solve a system of inequalities. For simplicity, we illustrate the argument for a two-locus system. Consider the order $w_{10} > w_{01} > w_{11} > w_{00}$. We can assume that

$$w_{00} = 1, w_{10} = 1 + a_1, w_{01} = 1 + a_2, \text{ where } a_1, a_2 > 0.$$

Additive fitness would imply $w_{11} = 1 + a_1 + a_2$, and then the rank order implies $1 + a_1 > 1 + a_1 + a_2$,

which is a contradiction. It follows that the rank order is incompatible with additive fitness. In general, a rank order combined with an additive assumption determines a system of linear inequalities. The rank order is compatible with additive fitness exactly if the system has a solution. It is not difficult to find software for solving such a system of inequalities. In particular any software for solving linear programming problems can be used.

The case $n \leq 3$ was discussed in the main text. For $n = 4$, one can verify computationally that there exist 14 rank orders (up to equivalence) with no order perturbations. As outlined, one can verify that all 14 rank orders are compatible with additive fitness. For $n = 5$, the analogous statement is not true, which was first shown by *Kraft et al. (1959)*. An explicit counterexample is given in *Maclagan (1998)*. This author studies Boolean term orders, in our terminology perturbation free rank orders, and refers to an order as being coherent if it is compatible with additive fitness. (The counterexample is described in slightly different notation, and the translation to our notation is

$$\emptyset \mapsto 00000, \{1\} \mapsto 10000, \{2\} \mapsto 01000, \{3\} \mapsto 00100, \{1,2\} \mapsto 11000, \{1,2,3\} \mapsto 11100,$$

and analogously.)

As explained, in principle, one can check whether a given rank order is compatible with additive fitness. However, the theoretical understanding of rank orders and additive fitness is still limited.

## Rank orders and statistical significance

As noted in the main text, rank-order methods are sensitive for measurement errors. A sufficient number of tests is necessary for reliable results, and elementary probability theory provides some guidance. If, for instance, genotype $g$ beats genotype $g'$ in 9 out of 10 comparisons, then it is reasonable to reject a null hypothesis of equal fitness.

Research on inferring rank orders from pairwise comparisons has a long history because of applications to sports and games (*Wauthier et al., 2013*; *Boyd and Silk, 1983*; *Albers and de Vries, 2001*; *Bradley and Terry, 1952*; *Thurstone, 1927*). However, statistical significance for rank orders has received considerably less attention, and it would be interesting to develop more theory on the topic.

**Simulations**. The upper graph in *Figure 3* was obtained by assuming additive fitness, a fitness decrease by 0.01 for each mutation, and errors sampled from a Gaussian distribution, where the magnitude of the errors is a value on the list 0.0010, 0.0011, 0.0012, . . ., 0.01. The assumptions were the same for the lower graph, except that a Student's t-distribution with df = 3 was used.

## Acknowledgements

We are grateful to Casey Aguilar-Gervase, Tonia Bell, Payal Dudheida and David Dunleavy for their studies on rank orders of genotypes for four-locus systems, and to Ethan Christensen for his work on antimalarial drug resistance.

## Additional information

### Funding
No external funding was received for this work

### Author contributions
Kristina Crona, Conceptualization, Formal analysis, Methodology

### Author ORCIDs
Kristina Crona https://orcid.org/0000-0003-1819-474X

### Decision letter and Author response
Decision letter https://doi.org/10.7554/eLife.51004.sa1
Author response https://doi.org/10.7554/eLife.51004.sa2

## Additional files

### Supplementary files
• Transparent reporting form

### Data availability
All data generated or analyzed during this study are included in the manuscript and supporting files.

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
