## [Decision Letter]

**Acceptance summary:**

This work extends a previous paper in *eLife*, where it was shown how the sign of certain epistatic interaction coefficients can be inferred from partial rank orders of fitness values. The purpose of the manuscript is to clarify the relation of this inference with the concept of sign epistasis; Crona et al., 2017, had shown that sign epistasis was a sufficient, but not necessary, condition for rank order to be informative about at least some signed interactions. Here the range of informative rank orders is extended by introducing the concept of rectangular perturbations, which generalizes the concept of sign epistasis by asking for the background dependence of the effect of mutational events that modify several loci at once. In the revision, the presentation was improved so that it has become more accessible for (evolutionary) biologists with a non-mathematical background.

**Decision letter after peer review:**

Thank you for submitting your article “Rank orders and signed interactions in evolutionary biology” for consideration by *eLife*. Your article has been reviewed by three peer reviewers, including Joachim Krug as the Reviewing Editor and Reviewer #1, and the evaluation has been overseen by Diethard Tautz as the Senior Editor. The following individual involved in review of your submission has agreed to reveal their identity: Luca Ferretti (Reviewer #3).

The reviewers have discussed the reviews with one another and the Reviewing Editor has drafted this decision to help you prepare a revised submission.

Summary:

This submission refers to the authors' previous *eLife* publication, Crona et al., 2017, where it was shown how the sign of certain epistatic interaction coefficients can be inferred from partial rank orders of fitness values. The purpose of the manuscript is to clarify the relation of this inference with the concept of sign epistasis. Crona et al., 2017 showed that sign epistasis is a sufficient condition for a rank order to be informative about at least some signed interactions, however it is not a necessary condition. Here the range of informative rank orders is extended by introducing the concept of rectangular perturbations, which generalizes the concept of sign epistasis by asking for the background dependence of the effect of mutational events that modify several loci at once. The exploratory application of the new concept to two empirical data sets suggests that it does give access to additional information about the topography of the landscape.

Overall, the reviewers found that, while the manuscript contains some interesting ideas and results that are worth to be reported in the form of a Research Advance, substantial revisions are necessary. Most importantly, the results should be presented in a way that is reasonably accessible for (evolutionary) biologists with a non-mathematical background, and the presentation also needs to be more systematic. A detailed list of the issues that should be addressed is given below.

Essential revisions:

1) All three reviewers found that the manuscript was lacking in clarity and context. Specifically, they felt that both the motivating question of the manuscript (as it arises from the original *eLife* publication of Krona et al., 2017), and the final answer that is achieved need to be spelled out more clearly and concisely. Moreover, it was not clearly stated how the new method of epistasis analysis goes beyond more established approaches, and in what sense it “add(s) precision to the analysis”.

2) Reviewers #2 and #3 addressed the generalizability of the results. One was concerned that the scaling of the number of perturbations as 6^n^ would limit the practical applicability of the method, and both asked whether and how the approach is generalizable to multi-allelic sequence spaces.

3) Another issue related to the practical application of the approach concerns the effect of measurement error. If measurement errors are not considered and there are many values close to each other, then the ranks could change significantly and the rank order could be heavily influenced by random noise, generating spuriously inferred interactions. The author states that “the results are not sensitive for a few false positives or negatives”, but this statement is not supported in the text aside from its assertion. This must be addressed (maybe through simulation?) before publication.

4) Reviewers #1 and #3 were concerned that the text assumes too much previous knowledge about fitness landscapes and specifically the polytope theory of Beerenwinkel et al., 2007, on the part of the reader. One reviewer found that the introduction of the “circuits from polytope theory” is not helpful and probably incomprehensible to readers who are not already familiar with the concept, and the other wondered how to understand remark 2.3 about “the class of circuits with non-zero coefficients for exactly four elements” when the word “element” has never been used before in the text. Perhaps a formal definition of circuits can be avoided by simply saying that circuits are linear combinations of the 4 genotypic fitness values that with coefficients +/-1 arranged in such a way that the expression vanishes when fitness is additive?

5) Reviewers #1 and #2 found that the discussion of the two rank orders that are compatible with additive fitness in the n=3 case in “Example 2.4” requires further clarification. What is “relabeling” in this context? And is it correct that the allowed orders are those where either the fitness is monotonic in the number of mutations, or at most one pair of genotypes violates this monotonicity, and this is a pair of genotypes with 1 and 2 mutations?

6) Reviewers #2 and #3 addressed the comparison to empirical data. Reviewer #2 was not convinced that the proposed method allows one to gain “new insights” beyond previous findings, and asks for further specification of the unique contribution of the method. Reviewer #3 suggests to place the results into the context of previous analyses of the same empirical landscapes [Ferretti et al., JTB 2016; Blanquart and Bataillon, Genetics 2016]. Moreover, it was not clear whether the proposed method (as applied to empirical landscapes) singles out the wild type sequence, or whether it treats all genotypes on the same footing.

7) Reviewer #1 requested further clarification of the precise relation between sign epistasis and signed interactions for different values of n (“Example 2.4”). It appears that the information contained in the manuscript regarding this point can be summarized as follows:

i) For n=2 loci instances of sign epistasis and signed interaction coincide.

ii) For n=3 and 4, any signed interaction corresponds either to an instance of sign epistasis or to a rectangular perturbation of higher order. If neither of these are present, the rank order is compatible with additive fitness.

iii) For more than 4 loci, the characterization by rectangular perturbations is insufficient to decide whether a rank order is compatible with additive fitness.

Statement (i) is obvious and well known, and the case n=3 has apparently been covered by the author through exhaustive enumeration. However, it is clear neither where the statement for n=4 comes from, nor in what sense the paper of Kraft et al., 1959, is relevant in the present context. Is there a characterization beyond rectangular perturbations that would apply for n > 4, and what kind of objects would it involve?

[Editors' note: further revisions were suggested prior to acceptance, as described below.]

Thank you for resubmitting your work entitled "Rank orders and signed interactions in evolutionary biology" for further consideration by *eLife*. Your revised article has been evaluated by Diethard Tautz (Senior Editor) and a Reviewing Editor.

The manuscript has been improved but there are some remaining issues that need to be addressed before acceptance, as outlined below:

Major issue

The Results section does not read well in its present form and needs to be reorganized. Presently the section begins with a partial explanation of the theory, followed by the discussion of experimental and hypothetical examples, and the complete account of the theory is found at the end. As a consequence, several concepts and definitions (e.g., the expression for the interaction u and the conditions for sign epistasis and rectangular perturbations) are introduced twice. I suggest to subdivide the section into subsections to improve readability, and to present the theory fully in the first subsection. I would also suggest to emphasize the simple geometric meaning of the various expressions in terms of the squares and rectangles in Figures 1 and 2: The circuits in the set C are obtained by going around one of the square or rectangle and giving the genotype fitness values alternating signs along the way. Similarly the conditions for sign epistasis and higher order perturbations amount to comparing fitness differences (= directions of arrows) on two opposite sides of a square/rectangle. Regarding the hypothetical examples, there is something wrong with landscape B, since the genotypes with 3 1's are not peaks. Also, I did not find the discussion of the epistasis measure of Ferretti et al. very illuminating (and I did not understand why it is called \lambda instead of \gamma as in the original paper). It seems to me that both the hypothetical examples and the discussion of Ferretti et al. could be omitted without losing much content.

---

## [Author Response]

Essential revisions:1) All three reviewers found that the manuscript was lacking in clarity and context. Specifically, they felt that both the motivating question of the manuscript (as it arises from the original eLife publication of Krona et al., 2017), and the final answer that is achieved need to be spelled out more clearly and concisely. Moreover, it was not clearly stated how the new method of epistasis analysis goes beyond more established approaches, and in what sense it "add(s) precision to the analysis".

In line with the reviewers suggestions, I have expanded text in the Introduction on how the results connect to open problems from Crona et al., 2017, added a separate Discussion section that summarizes conclusions, and also added a discussion about measurement errors to the Materials and method section. Most important, I have provided more evidence that rectangular perturbation goes beyond established approaches.

A clarification regarding “added precision” is probably necessary. If the available information is complete fitness measurements of the preferred kind (say Wrightian fitness), then rank order methods cannot add precision. There are good reasons to consider rank orders anyway. Observations such as many peaks, prevalent sign epistasis and few trajectories to the global peak, provide intuition for the evolutionary potential.

The “added precision” refers to how rectangular perturbations perform as compared to other rank order based concepts, such as summary statistics on sign epistasis, accessible mutational trajectories, peaks or fitness graphs:

A) The new method has the ability to detect epistasis for landscapes, also if there is no sign epistasis. An empirical example has been added as proof of principle.

B) Similarly, the new method reveals evolutionary important differences for landscapes that cannot be distinguished by frequently used methods. Explicit examples have been added.

C) In addition, the revised manuscripts describes a case where one can apply rectangular perturbations for handling a problem with measurement errors in sign epistasis data.

Note: One example from the original manuscript (on *Aspergillus Niger*) was deleted, since the added examples probably are more instructive. From the other original example (on malaria), the more complete statistics on rectangular perturbations that includes replacements 00 → 11, seems more informative than statistics of sign epistasis alone. In general, it seems rather obvious that checking the effects of replacements 00 → 11, in addition to 0 → 1 (sign epistasis) is meaningful, especially since no effort is required.

2) Reviewers #2 and #3 addressed the generalizability of the results. One was concerned that the scaling of the number of perturbations as 6^n^ would limit the practical applicability of the method, and both asked whether and how the approach is generalizable to multi-allelic sequence spaces.

In many cases it make sense to consider selected perturbations, depending on context. For very large systems, one can also sample genotypes and sets of loci and check for perturbations. (For a collection of [mostly] deleterious mutations, it would be interesting to check if replacements at several loci sometimes increase fitness.)

Rectangular perturbations can be defined for multi-allelic sequence spaces. However, a lot of theory on epistasis for biallelic n-locus systems (Walsh-coefficients, applications of the Fourier transform, fitness graphs) cannot easily be generalized to the multi-allelic case. Foundational work would be necessary for extending theoretical results to the multi-allelic case.

3) Another issue related to the practical application of the approach concerns the effect of measurement error. If measurement errors are not considered and there are many values close to each other, then the ranks could change significantly and the rank order could be heavily influenced by random noise, generating spuriously inferred interactions. The author states that "the results are not sensitive for a few false positives or negatives", but this statement is not supported in the text aside from its assertion. This must be addressed (maybe through simulation?) before publication.

I think the claim: “the results are not sensitive for a few false positives or negatives” caused some confusion (what I intended to say was merely that the conclusion for this particular example would not change because of a small number of false positives/negatives). I have deleted the claim and added a discussion about measurement errors to the Materials and methods section. In addition, the revised manuscripts describes a case where one can apply rectangular perturbations for handling a problem with measurement errors for sign epistasis data (Figure 3 in the revised manuscript shows a related simulation).

4) Reviewers #1 and #3 were concerned that the text assumes too much previous knowledge about fitness landscapes and specifically the polytope theory of Beerenwinkel et al., 2007, on the part of the reader. One reviewer found that the introduction of the “circuits from polytope theory” is not helpful and probably incomprehensible to readers who are not already familiar with the concept, and the other wondered how to understand remark 2.3 about “the class of circuits with non-zero coefficients for exactly four elements” when the word “element” has never been used before in the text. Perhaps a formal definition of circuits can be avoided by simply saying that circuits are linear combinations of the 4 genotypic fitness values that with coefficients +/-1 arranged in such a way that the expression vanishes when fitness is additive?

The manuscript is supposed to be much easier than Beerenwinkel et al., 2007, and several other cited articles. Figure 2 is intended to explain everything a reader needs to know for understanding the main idea. I have followed the reviewers suggestion and replaced my original definition of the circuits we use with a brief description, and moved the discussion about general circuits to the Materials and methods section (some readers will appreciate the full context).

5) Reviewers #1 and #2 found that the discussion of the two rank orders that are compatible with additive fitness in the n=3 case in “Example 2.4” requires further clarification. What is "relabeling" in this context? And is it correct that the allowed orders are those where either the fitness is monotonic in the number of mutations, or at most one pair of genotypes violates this monotonicity, and this is a pair of genotypes with 1 and 2 mutations?

In brief, a pair of rank order differ by labels only, if there exists a cube isomorphism that induces a map between the rank orders. A detailed description has been added to the Materials and methods section.

Yes, the claim is correct. For clarity, if we assume that ω_000_ < ω_g_ for each *g* ≠ 000, then additivity would imply that ω _111_> ω g for all *g* ≠ 111. If we also assume ω _100_> ω _010_> ω _001_, then the additivity assumption will impose further conditions on the rank order. A remaining question is whether ω 100 > ω _011_or ω _100_< ω _011_. If that question has been answered, the order is completely determined.

6) Reviewers #2 and #3 addressed the comparison to empirical data. Reviewer #2 was not convinced that the proposed method allows one to gain “new insights” beyond previous findings, and asks for further specification of the unique contribution of the method. Reviewer #3 suggests to place the results into the context of prevous analyses of the same empirical landscapes [Ferretti et al., JTB 2016; Blanquart and Bataillon, Genetics 2016]. Moreover, it was not clear whether the proposed method (as applied to empirical landscapes) singles out the wild type sequence, or whether it treats all genotypes on the same footing.

My response to comment 1 answers this question as well. In particular, an analysis of sign epistasis alone does neither have the same ability to rule out additive fitness, nor to capture global aspects, as a complete analysis of rectangular perturbation has.

For clarity, I have rephrased the text in one place in the revised manuscript. The new text is: “As proof of principle, we demonstrate that the large rectangles give new insights for two empirical studies (as compared to an analysis of sign epistasis)”.

I have added a remark about measures λ and λ* from Ferrettti et al., 2016. The proposed methods does not single out the wild type sequence. If the genotypes of highest and lowest fitness have maximal distance, it would be perhaps be most natural to assign the zero-string label to the genotype of lowest fitness (regardless if wild-type or not).

7) Reviewer #1 requested further clarification of the precise relation between sign epistasis and signed interactions for different values of n (“Example 2.4”). It appears that the information contained in the manuscript regarding this point can be summarized as follows:i) For n=2 loci instances of sign epistasis and signed interaction coincide.ii) For n=3 and 4, any signed interaction corresponds either to an instance of sign epistasis or to a rectangular perturbation of higher order. If neither of these are present, the rank order is compatible with additive fitness.iii) For more than 4 loci, the characterization by rectangular perturbations is insufficient to decide whether a rank order is compatible with additive fitness.Statement (i) is obvious and well known, and the case n=3 has apparently been covered by the author through exhaustive enumeration. However, it is clear neither where the statement for n=4 comes from, nor in what sense the paper of Kraft et al., 1959, is relevant in the present context. Is there a characterization beyond rectangular perturbations that would apply for n > 4, and what kind of objects would it involve?

If the rank order implies that υ > 0 for n = 2, we do not know whether or not the system has reciprocal sign epistasis. All we know is that the system has sign epistasis. In other words, signed circuits do not reveal complete information about sign epistasis, not even for n = 2. Similarly, a signed circuit interaction for c ε C tells us that there are one or two perturbations corresponding to c (differently expressed, one or two pairs of parallel arrows in the corresponding rectangle disagree). Note that potential rectangular perturbations are exactly twice as many as the number of circuits in C.

As for additivity, checking rectangular perturbations is sufficient for n < 4. The case n = 4 is reasonably straight forward from a computational point of view (the Materials and methods section describes how one can verify the claim). The reviewer’s last question is very interesting. Some relevant results are in Maclagan, 1999, and work that cites the paper (in particular detailed statistics for n = 5). However, the theoretical understanding for rank orders and additivity is limited. Because of the questions I have made several changes:

a) added explanations and clarifications for n = 3; 4.

b) In addition to my reference to Kraft et al., 1969, a reference to Maclagan, 1999, has been added. Maclagan gives an explicit counterexample (which is probably easier to understand), and I have provided a “dictionary” that explains concepts and notation in the paper so that an interested reader can check.

c) I have described how one can determine if a given rank order is compatible with additivity in the Materials and methods section.

[Editors' note: further revisions were suggested prior to acceptance, as described below.]

The manuscript has been improved but there are some remaining issues that need to be addressed before acceptance, as outlined below:Major issueThe Results section does not read well in its present form and needs to be reorganized. Presently the section begins with a partial explanation of the theory, followed by the discussion of experimental and hypothetical examples, and the complete account of the theory is found at the end. As a consequence, several concepts and definitions (e.g., the expression for the interaction u and the conditions for sign epistasis and rectangular perturbations) are introduced twice. I suggest to subdivide the section into subsections to improve readability, and to present the theory fully in the first subsection. I would also suggest to emphasize the simple geometric meaning of the various expressions in terms of the squares and rectangles in Figures 1 and 2: The circuits in the set C are obtained by going around one of the square or rectangle and giving the genotype fitness values alternating signs along the way. Similarly the conditions for sign epistasis and higher order perturbations amount to comparing fitness differences (= directions of arrows) on two opposite sides of a square/rectangle. Regarding the hypothetical examples, there is something wrong with landscape B, since the genotypes with 3 1's are not peaks. Also, I did not find the discussion of the epistasis measure of Ferretti et al. very illuminating (and I did not understand why it is called \lambda instead of \gamma as in the original paper). It seems to me that both the hypothetical examples and the discussion of Ferretti et al. could be omitted without losing much content.

I have restructured the result section accordingly. However, there are still some “repeats” of concepts, because I first discuss the two-locus case in some detail, and then (immediately after in the revised version) the general case, so as not to overwhelm the reader.

I have removed the hypothetical examples and the discussion of the measures introduced in Ferretti et al., 2016, in line with the suggestions. (My reason for adding the examples in the first place was that I thought reviewers wanted more evidence for that the proposed method differs from checking for sign epistasis, but I don’t insist on the examples.)

I have also added the clarifications requested (see below).

I would also suggest to emphasize the simple geometric meaning of the various expressions in terms of the squares and rectangles in Figures 1 and 2: The circuits in the set C are obtained by going around one of the square or rectangle and giving the genotype fitness values alternating signs along the way.

A similar text has been added after the definition of rectangular perturbations.

Similarly the conditions for sign epistasis and higher order perturbations amount to comparing fitness differences (= directions of arrows) on two opposite sides of a square/rectangle.

A similar comment has been added right before the remarks in the result section.